# Contingent Synergistic Interactions between Non-Coding RNAs and DNA-Modifying Enzymes in Myelodysplastic Syndromes

**DOI:** 10.3390/ijms232416069

**Published:** 2022-12-16

**Authors:** Argiris Symeonidis, Theodora Chatzilygeroudi, Vasiliki Chondrou, Argyro Sgourou

**Affiliations:** 1Hematology Division & Stem Cell Transplantation Unit, Department of Internal Medicine, University Hospital of Patras, 26504 Patras, Greece; 2Medical School University of Patras, University Campus, 26500 Patras, Greece; 3Biology Laboratory, School of Science and Technology, Hellenic Open University, 26335 Patras, Greece

**Keywords:** myelodysplastic syndromes, acute myeloid leukemia, aberrant DNA methylation, altered ncRNA expression, epigenetic landscape

## Abstract

Myelodysplastic syndromes (MDS) are a heterogeneous group of clonal hematopoietic stem cell disorders with maturation and differentiation defects exhibiting morphological dysplasia in one or more hematopoietic cell lineages. They are associated with peripheral blood cytopenias and by increased risk for progression into acute myelogenous leukemia. Among their multifactorial pathogenesis, age-related epigenetic instability and the error-rate DNA methylation maintenance have been recognized as critical factors for both the initial steps of their pathogenesis and for disease progression. Although lower-risk MDS is associated with an inflammatory bone marrow microenvironment, higher-risk disease is delineated by immunosuppression and clonal expansion. “Epigenetics” is a multidimensional level of gene regulation that determines the specific gene networks expressed in tissues under physiological conditions and guides appropriate chromatin rearrangements upon influence of environmental stimulation. Regulation of this level consists of biochemical modifications in amino acid residues of the histone proteins’ N-terminal tails and their concomitant effects on chromatin structure, DNA methylation patterns in CpG dinucleotides and the tissue-specific non-coding RNAs repertoire, which are directed against various gene targets. The role of epigenetic modifications is widely recognized as pivotal both in gene expression control and differential molecular response to drug therapies in humans. Insights to the potential of synergistic cooperations of epigenetic mechanisms provide new avenues for treatment development to comfort human diseases with a known epigenetic shift, such as MDS. Hypomethylating agents (HMAs), such as epigenetic modulating drugs, have been widely used in the past years as first line treatment for elderly higher-risk MDS patients; however, just half of them respond to therapy and are benefited. Rational outcome predictors following epigenetic therapy in MDS and biomarkers associated with disease relapse are of high importance to improve our efforts in developing patient-tailored clinical approaches.

## 1. Introduction

Among the complex and multifactorial pathophysiology of MDS, hypermethylation of several genetic loci plays a dominant role in all phases of the disease, from the early steps of generation of a clonal cell population until disease evolution to AML. Since MDS is heterogeneous disease, identification of easily recognized prognostic factors is of tremendous importance, and to this point the international prognostic scoring system (IPSS) or its revised form (IPSS-R) represents standard and useful tools for the assessment of risk categorization and prognosis, taking into account cytogenetics, the number and severity of cytopenias and bone marrow blasts percentage [1,2,3].

The dominant role in the initial pathogenesis of MDS has various somatic mutations, occurring at genetic loci with essential role in growth, development and differentiation of hematopoietic cells, as well as in epigenetic alterations, since MDS patients display abnormal hypermethylation patterns across many genomic regions [4,5,6,7]. Commonly mutated genetic loci in MDS which are implicated in epigenetics include DNA methyltransferases *DNMT1* and mainly *DNMT3A*, the family of methyl-cytosine dioxygenases *TET1-3* and the isocitrate dehydrogenases *IDH1-2*, contributing to the generation of aberrant methylation/demethylation genomic imprints [8,9,10]. DNMTs catalyze the reaction of cytosine methylation on DNA CpG dinucleotides, while the family of TET enzymes converts 5-methylcytocine (5mC) to 5-hydroxy-methylcytocine (5hmC) and other downstream oxidative products, which leads to DNA demethylation in daughter cells due to the inability of DNMT1 to recognize the modification from 5mC to 5hmC [11]. Isocitrate dehydrogenases catalyze the oxidative decarboxylation of isocitrate to 2-oxoglutarate. Mutated *IDH1* catalyzes the synthesis of 2-hydroxyglutarate, which inhibits alpha-ketoglutarate dependent enzymes such as the TETs and the histone demethylases, which indirectly leads to an elevation in 5mC levels within genome and a hypermethylated phenotype [12]. These enzyme families have been associated with harmful pathogenetic background, often acquired with advancing age and they are well known to be involved in disorders, such as MDS and various cancers of epithelial origin. Currently, the enhancement or silencing of other gene loci encoding for non-coding RNAs (ncRNAs) have also highlighted their role in the onset and course of MDS, as well as in the response to various treatment approaches, applied to these patients and in the risk to pathophysiology of AML progression [13] Differential expression of long ncRNAs appears as potential biomarkers for disease progression [14,15]. Moreover, deviating expression profiles of other ncRNAs, such as miRNAs (micro-RNAs), are detected in plasma or in bone marrow mononuclear cells during transformation of MDS to AML or in treatment intervals with hypomethylating agents [16,17,18].

Hypomethylating agents (HMAs), mainly azacytidine (AZA) and decitabine (DAC), used in clinical practice, are primarily considered inhibitors of DNA methyltransferases (DNMTs), mostly of DNMT1 [19]. AZA and DAC are both chemical nucleoside analogs of cytidine with identical ring structure, thus acting also as antimetabolites inside the rapidly proliferating immature clonal cells. These agents, after incorporation into DNA and/or RNA of highly proliferating cells, mainly induce DNMT1 depletion and global DNA hypomethylation. However, their function is not equivalent and distinctly different effects have been reported for their specific mode of action, with AZA having a greater effect on reduction of cell viability and total protein synthesis and also, restoration of onco-suppressing gene’s expression [20].

Efforts to understand in depth the underlying pathogenetic mechanisms of MDS have not attained much therapeutic progress in recent years. HMAs remain for almost two decades the mainstay of treatment for non-transplant eligible patients with higher-risk MDS (HR-MDS) and AML, despite the fact that only 30–50% of them may achieve a substantial response [21,22] and is also used as bridging treatment for several patients, candidates for allogeneic stem cell transplantation (Allo-SCT). Despite the long-term use of HMAs, an association of demethylating effect on specific DNA hypermethylated loci with particular clinical patterns of response, remains yet to be shown, while their broader mode of action in gene expression regulation is still unclear. According to the existing clinical experience, median overall survival with HMA monotherapy is ∼18–20 months for HR-MDS patients [23] and less than a year for those with AML [21]. This observation implies that additional mechanisms, beyond specific demethylation genomic events are developed, ascribing to clonal cells a biologically more aggressive/leukemic phenotype. Thus, several classes of novel agents, promising to act synergistically with HMAs are emerging, to overcome the generation of resistance. RNA-based therapeutic agents are still in pre-clinical investigation and only a few of them have entered the clinical stage to evaluate their value in practice. A novel RNA treatment approach aiming to immunize MDS and AML patients against bone marrow clonal cells, is the use of autologous dendritic cells (DCs) loaded with specifically synthetic mRNA molecules encoding tumor-associated antigens (including growth factors and unique antigens preferentially expressed in cancer cells), which is in the clinical trial stage (NCT03083054).

Given that the majority of MDS patients achieve relatively short-term responses and the inevitable development of resistance to HMAs, the discovery of new molecular, disease-specific or patient-specific biological prognostic factors, besides the complete understanding of the molecular mechanisms underlying the HMAs efficacy, are of high clinical importance. In this context, examples from the epigenetic field and possible interrelated actions between non-coding RNA molecules and enzymes affecting genome methylation/demethylation status are discussed in this review.

## 2. The Course of Aberrant DNA Methylation Patterns in MDS

DNA methylation is an essential epigenetic mechanism, playing very important role in the regulation of gametogenesis and zygote development in mammals (orchestrating the rounds of erasing and re-methylating of maternal and paternal derived chromosomes), imprinting and tissue-specific gene’s expression, and preservation of DNA hypermethylation as a suppressing mechanism for the locomotion of repetitive DNA elements [24,25,26]. Herein, enzymes responsible for epigenetic gene reprogramming are considered as delicate sensors of environmental signaling leading to cell responses and differential gene expression profiles. In particular, the epigenetic mark created by a methyl group covalent transfer, derived from the S-adenosyl methionine (SAM), to the 5′C position of a cytosine (C) either in scattered CpG dinucleotides across DNA sequences or in CpG islands, is catalyzed by DNMTs. By a rough classification DNMT1 is considered as the maintenance DNMT, which targets the hemi-methylated double stranded (ds)DNA to preserve the DNA methylation patterns post DNA replication and DNMT3A/DNMT3B as the de novo methylation enzymes, reacting with hemi-methylated and unmethylated dsDNA [27,28]. Contrary to DNMT1, responsible for the inherited DNA methylation, DNMT3A and DNMT3B are normally expressed and act during embryonic development or gametogenesis, independently from DNA replication. These enzymes share identical enzymatic centers at carboxyl-ends and different sequence patterns across N-ends that serve for communication with other molecules and for recognition and binding to DNA [29]. DNA recognition by DNMT3A and DNMT3B, apart from the N-terminal PWWP domain (‘Pro-Trp-Trp-Pro’ core amino acid sequence) by which bind to DNA, depends also on flanking sequences neighboring CpG targets. DNMT3B shows a significant preference for CpG methylation in a TACG (G/A) context [30], while other research groups have predicted that DNMT3A and DNMT3B preferentially methylate some representative sequences, more frequently found in naturally over-methylated genes [31,32] or that DNMTs in general, favor recruitment at DNA repair sites [33]. Furthermore, it has been reported that methylation of the Caspase-8 (*CASP8)* gene promoter in glioma cells is governed by both DNMT1 and DNMT3A, indicating a potential collaboration and concerted action of DNMTs [34]. Specific recruitment assisted by transcription factors or chromatin remodeling complexes is another context supporting DNMTs’ symmetrical function on DNA methylation [35]. However, there are still some interesting questions, such as the broader DNA architecture and/or epigenetic landscape that allows DNMTs to bind to CpG target sites in preferable or cognate sites or whether the flanking sequence preferences adapt to specific biological targets of DNMTs. Within this frame of action, crosstalk of DNMTs with non-coding RNA species as potential guides to specific sites for methylation across genome, remains an issue deserving further investigation.

DNMT3A and DNMT3B are frequently associated with divergent de novo DNA methylation patterns and gene repression in many pathologies. A common observation is that Myelodysplastic Syndromes are developed under an aberrant epigenetic background [10]. Cell cycle regulators, apoptotic genes, and DNA repair genes are irregularly silenced through epigenetic modifications, promoting clonal dominance and expansion of abnormal hematopoietic stem cell, favoring gradual disease progression or in association with various other somatic mutations, the transformation to AML [36]. Figure 1 illustrates phenotypic and functional cell alterations in hematopoietic stem cell (HSC) compartment of bone marrow, associated with HR-MDS that guide disease progression to AML. In MDS-derived secondary AML, mutations among the *DNMTs* and the *TET* family of genes contributing to demethylating genome pathways are frequently identified, suggesting that aberrant epigenetic programming plays a crucial role in MDS progression [10].

Cytosine methylation in CpG dinucleotides is a critical epigenetic modification, although it can be reversible. DNA demethylation occurs via a stepwise oxidation of 5-methylcytosine that is catalyzed by the Ten-Eleven Translocation or methyl-cytosine dioxygenases (TET) enzymes. The first oxidation product in this process, 5-hydroxymethylcytosine (5hmC), has been shown to act as a unique epigenetic mark, which, in contrary to 5mC, is linked to transcriptional activation. 5hmC has been implicated in the activation of lineage-specific enhancers and in substantial cell processes [37]. The oxidation pathway generates also several other intermediates (i.e., formylcytosine and carboxylcytosine), that have their own distinct biological functions [38]. The 5hmC has a functional role in promoting gene expression during active demethylation, where conversion of 5mC to 5hmC by the TETs prevent the recognition of DNA sequences by repressive (Methyl-Binding) MBD-domain complexes and DNMT proteins that would typically be recruited to rich 5mC areas [39]. The issue of cytosine epigenetic modifications across DNA CpG rich sequences apparently provides another layer of regulation, beyond the canonical genetic codes.

Hypomethylating agents (HMAs) are mainly administered as treatment to HR-MDS patients with increased percentage of bone marrow blasts, carrying a high risk for AML development [40]. Among the most widely administered MDS treatment approaches, AZA and DAC, are considered inhibitors of DNMT1, which is almost completely depleted after HMA exposure, whereas DNMT3A is significantly less sensitive and DNMT3B seems completely resistant to HMAs [19]. DAC is tri-phosphorylated and is incorporated into newly synthesized DNA as a substitute for cytosine (antimetabolite), which pairs with guanine assisted by DNA-polymerase, in contrast, AZA is converted to ribonucleoside triphosphate and is incorporated into RNA, leading to inhibition of protein synthesis [41]. Investigation has shown that DNMT1 activity decreases faster than incorporation of HMAs into DNA [19,42,43] and DNMT1 depletion occurs even in the absence of DNA replication and cell division [44,45]. The DNMT1 depletion mechanism proposed, supporting these data, is that HMAs induce DNMT1 degradation in the nucleus via its rapid hyperphosphorylation by the protein kinase C delta (PKCd), followed by ubiquitination and finally leading to proteasomal degradation [19]. Although shared epigenetic mechanisms of action have been affiliated to both AZA and DAC, such as the DNMT1 depletion and global DNA hypomethylation, their function is not equivalent. Distinctive effects have been reported on many cellular responses, such as cell viability and gene expression [20]. In proliferating cells, inactivation of DMNT1 after starting treatment with HMAs, results in a persistent hemi-methylated DNA status delivered to next generations of daughter cells. However, HMAs’ reversible activity towards the abnormal epigenetic stem cell profiles and reactivation of aberrantly silenced genes is moderate to low [46]. Response to HMA therapy is not always easily predictable, since the required molecular genetic testing is not usually applied in routine clinical practice and most MDS patients who receive HMA therapy develop resistance to treatment over time and disease progression to AML [47].

In addition to DNMTs, other epigenetic components, such microRNAs (miRNAs) and small interfering RNAs (siRNAs) exhibit altered expression profiles in MDS patients following treatment with HMAs and during disease progression, while long non-coding RNAs (lncRNAs) are shown to direct the epigenetic mechanisms and chromatin remodeling complexes in many genetic loci influencing gene expression. These perspectives are discussed in the next sections.

## 3. Long Non-Coding RNAs and MicroRNA Species’ Involvement in MDS

Nuclear non-coding RNAs (ncRNAs), either long or short, are exempt from protein-encoding obligations. The inherent non-protein coding nature of these transcripts is documented by the absence of an Open Reading Frame (ORF) and further confirmed by computational and biochemical approaches. Although produced intra-cellularly, ncRNAs may even found in the extra-cellular space and body fluids as parts of the micro-vesicles and/or exosomes [48,49,50]. Among the ncRNA categories, lncRNAs are characterized by their extensive length of more than 200 nucleotides and are considered as products of the RNA polymerase II, partially spliced and often polyadenylated [51]. However, a proportion of human lncRNAs identified by RNA-seq are not RNA Polymerase II transcripts [52]. Non-polyadenylated lncRNAs are transcribed by RNA polymerase III or processed by RNase P cleavage of tRNA-like structures to generate a mature 3′ end or capped by snoRNP complexes at both ends or by forming circular structures [53,54,55,56]. Particularly in AML, several lncRNA species have been identified with evidently adverse roles in disease progression. *DANCR*-lncRNA is upregulated in leukemic stem cells obtained from AML patients, promoting stem cell prolonged survival and self-renewal capacity [57]. *LEF1-AS1, MALAT1* and *NEAT1* have also oncogenic potential in AML [14]. Another category of lncRNAs that includes *HOTTIP, UCA1* and *MEG3* tends to sponge tumor suppressor microRNAs, thus stimulating AML reproductive capacity [14]. On the contrary, *H22954*-lncRNA acts as a tumor suppressor molecule in AML inducing cell apoptosis [58].

Small ncRNAs of ~20–30 nucleotides (nt) in length, have also emerged as key players in various biological processes. In mammals, major classes of small RNAs are the miRNAs, the siRNAs and the Piwi-interacting RNAs (piRNAs) [59,60]. MiRNAs and siRNAs are processed from double-stranded (ds) RNA precursors by the intranuclear Double-Stranded RNA-Specific DROSHA enzyme (or through other, DROSHA-independent processing pathways) and subsequently by DICER enzyme, two RNase type III enzymes that catalyze ncRNAs hairpin structures dissociation to generate small dsRNAs. They also interact with members of the Argonaute (AGO) family and are considered to mediate gene expression at the transcriptional or post-transcriptional level in their mature single stranded conformation via complementarity with their targets (mainly mRNAs) [61]. SiRNA molecules with accompanying Argonaute-binding proteins compose the assembled parts of the RNA-induced transcriptional silencing (RITS) complex directing epigenetic chromatin modifications. Association of RITS with nascent RNA transcripts at target loci is stabilized by proximate binding to methylated histone H3 at lysine residue 9 (H3K9me) and guide heterochromatin formation and transcriptional silencing. RNA polymerase II (Pol II) is often linked to the whole process [62].

MDS patients are also characterized by altered profiles of circulating miRNAs in plasma and ‘exosomes’ (vehicles encapsulating DNA, proteins and RNA species) as well as in bone marrow mononuclear cells (BMMC), a mixed population of single nucleus cells including monocytes, lymphocytes, NK cells and hematopoietic stem cells. Several research groups have investigated variations in miRNA subpopulations during early or advanced stages of MDS and following the administration of hypomethylating agents or progression to AML, to find representative and informative biomarkers as potential prognostic tools (Table 1). Substantial results have shown that increased miR-196b-5p, miR-320c, miR-320d, miR-422a, miR-617, miR-181a, miR-222, miR-210 and let-7a or decreased miRNA-29b expression levels were correlated with poor prognosis of MDS and increased risk for AML development [16,17,63,64]. MiR-30b, miR-30e and miR-221 have been shown to be downregulated in MDS and have been implicated in jak-STAT signaling and Th17 cell differentiation pathways as well as in cytokine–cytokine receptor interactions [18]. Characteristic MDS cytogenetic changes have been correlated with unique miRNA expression profiles: trisomy 1 is accompanied by decrease in the relative expression level of miRNA-194-5p, del(5q) by substantial decrease of miRNA-378 and miRNA-146a and by miRNA-34a increase, chromosomal translocation t(p21;q23) by increased expression levels of miRNA-125b-1, whose gene is located close to the chromosome 11q23 breakpoint, and finally, trisomy 8 by elevation of miRNA-383 expression [65]. The methylation status of miRNA promoter regions has also been proved to play a particular role, towards permissive or disincentive transcription. Hypermethylation of miRNA-34b promoter was detected during AML transformation in MDS patients [16], while hypomethylation of other miRNA loci, such as let-7a-3 gene (member of the let-7 gene family which encodes for mature miRNA let-7a together with let-7a-1 and let-7a-2) and miRNA-124-3 contribute to transformation to AML and predict poor outcome for MDS patients [66,67]. The interrelationship between different epigenetic mechanisms deserves further investigation, in terms of decoding underlying regulatory components favoring the epigenetic instability in MDS and providing new insights towards novel treatments. Among cell pathways in which miRNAs are involved, the most common target genes are Cyclin-Dependent Kinase Inhibitor 1B (*CDKN1B*), Cyclin Dependent Kinase 6 (*CDK6*)*,* Mitogen-Activated Protein Kinase 1 (*MAPK1*)*,* E3 Ubiquitin Protein Ligase (*MDM2*), which is a Proto-Oncogene and Protein Kinase DNA-Activated Catalytic Subunit (*PRKDC*), targeted by at least four different miRNAs presented in Table 1. Both *CDKN1B* and *CDK6* control cell cycle G1 phase progression, *MAPK1* is an essential component of the MAP kinase signal transduction pathway, *MDM2* is implicated in cell cycle arrest and apoptosis via TP53 tumor suppressor protein and *PRKDC* is involved in double strand break repair and recombination. All cellular processes described emphasize the individual-tailored epigenetic variability, which is orchestrated by separate groups of epigenetic features to promote the disease manifestation and course.

## 4. Non-Coding RNA Species Cooperate with DNA Modifying Enzymes

Interaction of lncRNAs with genome and DNA binding proteins or chromatin remodeling complexes has been thoroughly assessed, with *XIST*-lncRNA being recognized as the most prominent example for the inactivation of one homologous X chromosome in females [68,69]. Another nuclear non-polyadenylated *CEBPA*-lncRNA originating from the CCAAT/Enhancer-Binding Protein Alpha (*CEBPA*) locus, regulates *CEBPA* methylation levels. The encoded protein (CEBPA) modulates expression of genes involved in cell cycle phase maintenance or transition and in body weight homeostasis. Mutations of this gene have been associated with AML [70]. The proposed underlying mechanism is that *CEBPA*-lncRNA interacts with DNA in a sequence specific manner and with DNMT1 protein preventing by this strategy the *CEBPA* methylation. Prevention of methylation leads to high level of *CEBPA*-mRNA transcription and suggests a novel regulatory mechanism of gene methylation governed by lncRNAs [71,72]. MDS patients exhibit modified levels of gene expression profiles including lncRNAs and other non-coding RNAs. The expression of several lncRNAs has been correlated with specific clinical and molecular features of MDS in various studies, which have demonstrated their importance as potential prognostic markers [73]. A DNMT1-associated lncRNA, the *DACOR1*-lncRNA (DNMT1-associated Colon Cancer Repressed 1), is down-regulated in colon tumors. Suppressed *DACOR1*-lncRNA affects unconstrained gene expression profiles in colon cancer cell lines via stimulation of genome-wide DNA demethylation [74], presenting a possible mechanism reflecting variances in response to HMAs among MDS patients. Analogous supportive evidence for interrelations between lncRNAs and enzymes of the epigenetic machinery derives from the long intergenic noncoding antisense RNA (*HOTAIR*), which preserves Homeobox A1 (*HOAX1*) gene’s methylation levels through induction of the evolutionarily conserved SET-domain-containing histone methyltransferase EZH2, the DNMT1 and DNTM3B in lung cancer [75,76]. Recently *HOTAIR*-lncRNA was shown to positively influence DNMT3B activity and increase methylation levels of Phosphatase and Tensin-Like protein’s gene *PTEN* and *HOAX5* in AML [77,78]. PTEN has been shown to be associated with HMA resistance in MDS and progression to AML, suggesting that it might represent a potential target for treatment in combination with HMAs [79]. Another homeobox (HOX) antisense transcript, *HOTAIRM1*-lncRNA, has been implicated to cell autophagy and enhanced cell proliferation in leukemic cells and has been associated with adverse prognosis in adult *NPM1* (nucleophosmin)-mutated AML [80,81]. In glioblastoma, *HOTAIRM1*-lncRNA acts by sequestering EZH2 and DNMTs from *HOXA1*, directing its up-regulation of expression [82]. Interestingly, *HOXB-AS3*-lncRNA, transcribed from the human *HOXB* cluster, is shown to recruit EZH2 to Dicer endoribonuclease’s promoter, an enzyme that cleaves double-stranded RNA and hairpins of the pre-micro and siRNAs [83], and also to be implicated in myeloid cell proliferation with adverse prognosis in AML and MDS [15]. Moreover, NR-104098-lncRNA inhibits AML proliferation and induces differentiation through repression *EZH2* transcription by recruiting E2F1 to its promoter [84], a transcription factor displaying high levels of expression in bone marrow of MDS [85]. Finally, upregulation of *H19*-lncRNA has been associated with adverse prognosis in MDS patients [86] and has been interconnected with DNMT3B function in breast cancer [87] by promoting reactivation of *BECN1* gene, a component of the phosphatidylinositol-3-kinase (PI3K) complex which mediates vesicle-trafficking processes, playing multiple roles in autophagy, also involved in MDS pathogenesis [88]. Along the DNA demethylation pathway, *MAGI2-AS3*-lncRNA inhibits the self-renewal of leukemic stem cells by promoting TET2-dependent DNA demethylation mechanism of the gene promoter encoding for the Leucine-Rich Repeat Protein’s (*LRIG1*) in AML, setting another example of ncRNA interaction with DNA modifying enzymes implicated in epigenome [89].

Functional characteristics and life cycles of identified lncRNAs linked to MDS should be underscored concerning their regulatory potential in DNA methylation and their universal action. Additionally, DNA methylation spreading restrictions depending on RNA inhibition is a cooperative epigenetic phenomenon and potentially represents a global way of inter-regulation between mechanisms, which are still considered distinct.

So far, there are few but well documented paradigms of interrelations between small ncRNA and proteins related to DNA or formation of DNA-ncRNA hybrids with regulatory functions. RNA molecules are known to impact DNA repair following damage, directly or indirectly by recruiting protein factors involved in DNA damage signaling and repair, particularly at DNA double strand breaks, considered as the most harmful DNA lesions. Production of DSB-induced small RNAs from sense and antisense sequences around DSB sites, has been obtained in human cells. DSB-induced small RNAs are associated with Argonaute (member of the RNA-induced silencing, RISC complex) and are required for the activation and efficient homologous recombination (HR) repair mechanism [90,91]. These phenomena indicate that ncRNAs intermediate the base-pairing with the damaged DNA or represent platforms responsible for DNA repair factors’ recruitment to the DSB, thus facilitating efficient DNA repair.

A few paradigms highlight the co-regulation of DNMTs, EZH2 and TET enzymes by miRNAs. MiRNA-21 and miRNA-148a are causing DNMT1 down-regulation in hematopoietic stem cells of patients with systemic lupus erythematosus [92]. MiRNA-21, which is recognized as a potential serum biomarker during epigenetic therapy in MDS [93], is also an up-regulator of *EZH2* in human lung cancer stem cells [94], whereas miRNA-148a has recently been recognized as a down-regulator of *DNMT1* in AML cells [95]. Additionally, miRNA-29b, down-regulates indirectly and directly *DNMT1* and *DNMT3A/3B* expression respectively in AML patients [96] and along with miRNA-29a cooperatively target DNMT3A/3B expression in lung cancer [97], while they are both down-regulated in AML [98]. MiR-185 and miR-152, act by suppressing DNMT1 activity in hepatocellular carcinoma cells [99] and prostate cancer respectively [100], promoting PTEN expression, also shown to be associated with HMA resistance in MDS [79]. Another DNMT1-inhibitor examined in prostate cancer cells [101], miRNA-342, is downregulated in AML [102]. In addition, miRNA-221, an inhibitor of DNTM3B in breast cancer cells [103], is significantly decreased in AML evolving from MDS [104]. Other epigenetic enzymes are also regulated by miRNAs, such as TET2, a direct target of miRNA-22, a non-coding RNA with known prognostic value in MDS treated with HMAs [105], shown to provoke MDS in mice [106]. Other miRNAs that interact with TET2 in macrophages is the let-7 family [107]. MiRNA-101, that is found to delay onset or progression to AML from MDS [108] and miRNA-124, which is a possible marker of response to DAC in MDS/AML [109], both repress cancer proliferation by EZH2 inhibition [110,111]. Additionally, miRNA-34b, directly targeting histone methyltransferases and deacetylases to inhibit prostate cancer [112], also found to inhibit cell viability and promote cell apoptosis in AML cell lines [113].

Under the enlightenment of these important but still restricted results, it is anticipated that non-coding RNAs could possibly mediate the accurate targeting of CpG sites across genome offering guidance to DNMTs via sequence complementarity (between DNA and RNA) or play the role of scaffolds that recruit other DNA-modifying enzymes. Non-coding RNAs with secondary loops, such as the inter-nuclear transcribed pre-miRNA clusters and lncRNAs govern the structural principles for DNA–RNA–protein interactions (Figure 2). Future perspectives for elucidation of these mechanisms in myelodysplastic syndromes described above, are presented in Table 2.

## 5. Conclusions and Future Perspectives

DNA methylation in several neoplastic disease states is altered as an aggregated consequence of the gradual loss of gene and non-coding RNA expression control (which is also age-related). Moreover, the decline of mitochondrial activity, the imbalanced macromolecules’ concentration and dysfunction, the potential accumulation of destructive misfolded proteins and the presence of reactive oxygen species (ROS) within cell are manifested as cellular stress. These cell conditions are linked to genetic instability and epigenetic remodeling effects across various genetic loci that lead to pathogenesis of more aggressive and heterogeneous phenotypes. However, cellular function is completely dependent on fidelity and correct transmission and translation of genetic information under physiological epigenetic environment. Research approaches on epigenetic phenomena are based on their advantageous reversible nature, provided that key features of vital importance occurring in the shifted epigenetic background are decoded and targeted for molecular interventions before they dominate.

Current data on the roles of lncRNAs, miRNAs and DNA methylation status in MDS suggest that have the potential to become prognostic and diagnostic tools for MDS. The same is also anticipated to be applicable for patients with sAML (MDS/AML), since this disease appears to be a more advanced stage of the same group of diseases, even classified among AML. MDS and AML risk stratification, although steadily directed to the incorporation of molecular factors towards treatment, still needs improvement. To this point, the intra- or extra-cell concentrations of various non-coding RNA species can be relevant to the treatment responses, and they could also be considered as therapeutic targets. Moreover, several non-coding RNAs may mutually cooperate with enzymes involved in epigenetic events across genome by either promoting recognition and uploading on specific DNA sequences or prevention of their function, even via stereochemical inhibition. These contingent synergistic interactions can be clarified through research and become new therapeutic targets since the ultimate definition of epigenetic regulation includes short- or long-term modifications with reversible properties.

Other advanced aspects that need to be assessed by further investigation include the actual time intervals that hypomethylating effect of HMAs remain indelible, the sustainability of the beneficial results of HMAs treatment, obtained when a favorable response is achieved and the identification of the best partners of HMAs which may increase response rate, deepen the response with clearance of leukemic stem cells and prolong the period of response. Furthermore, the tissue-specific neoplastic cell populations, which are affected from ΗΜΑ treatment and the chromosomal locations licensed to reverse an established epigenetic status of hypermethylation should be unraveled. Critical questions, such as whether the DNMTs target the same groups of CpGs in different cell types or how DNMTs selectively act on specific CpGs which become hypermethylated, while others remain rather unaffected, also need a deeper insight. Finally, another “dark spot” to be assessed is whether hypomethylation/demethylation of various tumor suppressor genes, such as cell-cycle transition inhibitors or TP53 or DNA repair genes and their re-activation are the keys to treatment response with HMAs. Alternatively, the consequent restoration of dysregulation of the immune system or other cellular pathways leading to beneficial outcomes for the MDS patients, needs to be further investigated and elucidated.

## Figures and Tables

**Figure 1 ijms-23-16069-f001:**
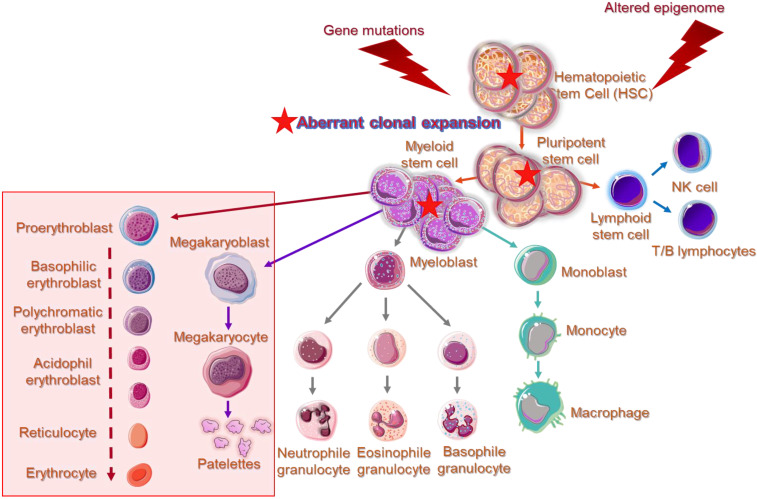
Overview of cells’ aberrations in epigenome of higher-risk MDS patients within the hematopoietic stem cell (HSC) compartment of bone marrow. Impaired gene expression implicated in maturation and differentiation of pluripotent stem cells give rise to myeloid lineage-committed cells showing further phenotypical as well as functional changes. MDS patients of the higher-risk group present significantly expanded granulocytic-monocytic progenitor cells and HSC compartments, while the megakaryocytic–erythroid progenitor cell population is severely reduced. Cellular and molecular changes are in line with the observed cytopenias and emergence of aberrant blast cells in the bone marrow and peripheral blood of higher-risk MDS patients. Aberrant cells characterized by the generation of multiple copies per cell, are indicated by a solid red star. Obstructed pathways of myelopoiesis and erythropoiesis are indicated within a transparent rectangle.

**Figure 2 ijms-23-16069-f002:**
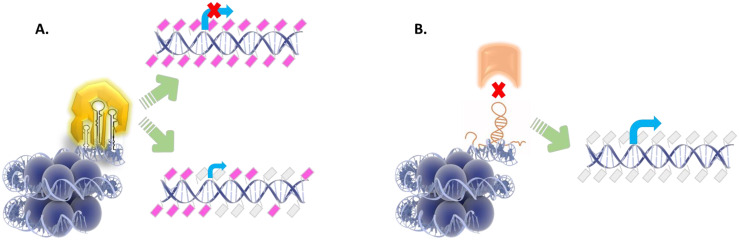
RNAs capable of adopting stem-loop structures exhibit the potential to associate with DNA modifying enzymes depending on RNA secondary structure. The basis of this preferential interaction and targeting of specific CpGs lengthwise genome is the complementarity of DNA-RNA sequences. (**A**). Immature pre-miRNA clusters may bind and provide a scaffold for either DNMTs or TET enzymes to DNA sites for de novo DNA methylation or demethylation events, respectively. DNA hypermethylation prevents transcription, while even partial demethylation promotes transcription albeit at minimal levels (lower). (**B**). DNA sequences occupied by lncRNAs may prevent enzymatic modifications to DNA by obstructing enzymes from binding, which results in an ongoing transcriptional activity.

**Table 1 ijms-23-16069-t001:** Up- and down-regulated miRNAs in MDS, related KEGG pathways and targeted human genes. Data were extracted via mirPath v.3 tool provided by DIANATOOLS (https://dianalab.e-ce.uth.gr/html/mirpathv3/).

Upregulated miRNAs	Tissue	Impaired KEGG Pathways	Gene Targets within KEGG Pathway
miR-196b-5p	Bone marrow	PI3K-Akt/TGFβ signaling	*BRAF*, *TGFBR1*, *E2F2*, *NRAS*, *AF1*, *CDKN1B*, *SMAD4*, *MYC*, *MAPK1*, *MDM2*
		ECM-receptor interaction	*ITGB1*, *ITGB8*, *THBS2*, *ITGA3*, *COL3A1*, *COL1A2*, *HMMR*
		cell cycle via MAPK signaling	*ESPL1*, *CCNB1*, *E2F2*, *CDK2*, *CND2*, *SMC3*, *CDKN1B*, *STAG1*, *YWHAB*, *SMAD4*, *SKP2*, *MYC*, *PLK1*, *MDM2*, *MCM3*
miR-320c, miR-320d	Bone marrow	TGFβ signaling	*PPP2CA*, *SMAD3*, *ACVR2B*, *SMAD7*, *MAPK1*
miR-422a	Bone marrow		*GNG12*, *MAPK1*, *GNG5*, *YAP1*
miR-617	Bone marrow	RNA transport, translation initiation	*XPO1*, *EIF5*, *EIF5B*, *EIF4G2*
miR-181a, further divided to miR-181a-2-3p, miR-181a-3p and miR-181a-5p	Bone marrow	Signaling pathways regulating pluripotency of stem cells	*IGF1R*, *HESX1*, *WNT8A*, *POU5F1B*, *ACVR2B*, *JAK3*
miR-222, further dived to miR-222-3p and miR-222-5p	Bone marrow	PI3K-Akt/TGFβ signaling	*SOS2*, *CDK4*, *E2F2*, *CRKL*, *RUNX1*, *CDKN1B*, *CDK6*, *E2F3*, *MYC*, *MAPK1*, *MDM2*
		cell cycle via MAPK signaling	*YWHAH*, *CDK4*, *E2F2*, *YWHAE*, *YWHAG*, *CDKN1B*, *WEE1*, *CDK1*, *CDK6*, *ATM*, *CDKN1C*, *E2F3*, *MYC*, *YWHAZ*, *CDC20*, *BUB1B*, *CDC27*, *PRKDC*, *MDM2*, *CDC25A*
miR-210, further divided to miR-210-3p and miR-210-5p	Bone marrow	No related pathway	
let-7a, further divided to let-7a-2-3p, let-7a-3p and let-7a-5p	Bone marrow	cell cycle via MAPK signaling	46 genes targeted/*CREBBP*, *CDK6*, *WEE1*, *MAP3K1*, *ELK4*, *MAX* and *TAOK1* confirmed by more than 1 lists
		FoxO, HIF-1 and PI3K-Akt signaling	Over than 100 genes targeted/*IRS2*, *CREBBP*, *BCL2L11*, *SLC2A1*, *TFRC*, *EFNA1*, *PRKAA1*, *COL4A1* and *CDK6* confirmed by more than 1 lists
miRNA-34a (further dived to miRNA-34a-3p and -5p) in del(5q) MDS		PI3K-Akt/TGFβ signaling	35 genes targeted
		cell cycle via MAPK signaling	Over than 100 genes targeted/*PRKDC, DUSP16* and *MCM7* confirmed by more than 1 lists
		Cell cycle and apoptosis via P53 signaling	34 genes targeted/*THBS1*, confirmed by more than 1 lists
miRNA-125b-1-3p in translocation (p21;q23) MDS	Plasma/bone marrow	No related pathway	
miRNA-383 (further divided to miR-383-3p and -5p) in trisomy 8 MDS	Bone marrow	No related pathway	
**downregulated miRNAs**			
miR-29b further divided to miR-29b-1-5, -2-5p and -3p	Bone marrow	PI3K-Akt/TGFβ signaling	63 genes targeted/*PKN2*, *SGK1*, *NRAS*, *CCND1*, *YWHAB*, *CDK6*, *MCL1* and *MDM2* confirmed by more than 1 lists
		cell cycle via MAPK signaling	33 genes targeted/*MDM2*, *SMC1A*, *YWHAB*, *CCND1*, *CDK6*, *HDAC1* and *WEE1* confirmed by more than 1 lists
		Cell cycle and apoptosis via P53 signaling	22 genes targeted/*CCND1*, *CDK6* and *MDM2* confirmed by more than 1 lists
		FoxO signaling	27 genes targeted/*GF1R*, *NRAS*, *STK4*, *SGK1*, *MDM2*, *CCND1* and *GABARAP* confirmed by more than 1 lists
miR-130b-5p	Plasma	Cell cycle and apoptosis via P53 signaling	*ZMAT3, CDK6, CHEK1, ATM, CCND1, MDM4,* *RRM2, SESN3, SERPINE1, PPM1D, MDM2*
miR-30e further divided to miR-30e-3p and -5p	Bone marrow	Cell cycle and apoptosis via TGFβ signaling	*FST*, *TGFBR1*,*SMAD2*, *THBS1*, *CUL1*, *INHBA*, *DCN*,*MYC*, *PPP2R1A*, *SMURF1*, *ZFYVE9*, *BMP2*, *SP1*, *EP300*, *SMAD7*, *E2F4*, *MAPK1*, *PPP2R1B*, *TGFBR2*, *BMPR2*, *RPS6KB1*, the underlined confirmed by more than 1 lists
		cell cycle via MAPK signaling	39 genes targeted/*MDM2*, *SMAD2*, *YWHAZ*, *PRKDC*, *STAG2*, *MYC*, *CDK6*, *CCNB1* and *E2F3* confirmed by more than 1 lists
		Cell cycle and apoptosis via P53 signaling	24 genes targeted/*MDM2*, *CDK6*, *CCNB1*, *CCNG1*, *CASP3* and *SIAH1* confirmed by more than 1 lists
		RNA transport, translation initiation	37 genes targeted/*XPO1*, *NCBP1* and *RANBP2* confirmed by more than 1 lists
miR-221 further divided to miR-221-3p and -5p	Plasma	mRNA surveillance pathway	21 genes targeted/*ACIN1* and *PABPC1* confirmed by more than 1 lists
		cell cycle via MAPK signaling	26 genes targeted/*CCND1* confirmed by more than 1 lists
		Cell cycle and apoptosis via P53 signaling	50 genes targeted/*EFNA1*, *CDKN1B*, *CCND1* and *ATF4* confirmed by more than 1 lists
miRNA-194-5p	Bone marrow	Ubiquitin mediated proteolysis	*WWP1*, *TRIP12*, *CBLB*, *BIRC6*, *SMURF1*, *DDB1*, *CDC23*, *RBX1*, *UBE2W*, *UBE2B*, *PIAS1*, *CUL4B*
miRNA-146a (further dived to miRNA-146a-3p and -5p) in del(5q) MDS	Peripheral blood mononuclear cells	cell cycle via MAPK signaling	*GSK3B*, *CCNB1*, *CDC25B*, *YWHAG*, *CDKN1B*, *SMAD4*, *RBL1*, *CDC23*, *CDKN1A*, *PRKDC, MDM2*, *ABL1*, *CDC25A*
miRNA-125b-1		No related pathway	

**Table 2 ijms-23-16069-t002:** Epigenetic enzymes coregulated by lncRNAs and miRNAs and implicated in MDS and AML.

Non-Coding RNA	Epigenetic Enzyme Targeted	Disease/Tissue of Investigation	Gene/Function Implicated	Possible Relation in MDS/AML Pathogenesis
*MAGI2-AS3*-lncRNA	↑TET2	AML	↑*LRIG1* [89]	-
*HOTAIR*-lncRNA	↑DNMT3B ↑EZH2, DNMT1/3B	AML SCLC	↓*PTEN* [77], ↓*HOXA5* [78] ↓*HOAX1* [76]	-
*HOXB-AS3*-lncRNA	↕EZH2	Liver cancer	↑*DICER1* [83]	Implicated in myeloid cell proliferation with adverse prognosis in AML and MDS [15]
*H19*-lncRNA	↨DNMT3B	BC	↑*BECN1*/ autophagy [87]	Associated with adverse prognosis in MDS patients [86] *BECN1* is also implicated in MDS autophagy [88]
*HOTAIRM1*-lncRNA	↨EZH2/DNMTs	GB	↑*HOAX1* [82]	Associated with leukemia cell autophagy and proliferation and adverse prognosis in *NPM1*-mutated AML [80,81]
NR-104098-lncRNA	↓EZH2 (via E2F1)	AML	Inhibits AML proliferation & Induces differentiation [84]	-
*CEBPA*-lncRNA	↨DNMT1	AML	↑*CEBPA* [71,72]	-
*DACOR1*-lncRNA	↨DNMT1	CRC	Whole genome demethylation [74]	MDS have abnormal methylation patterns across many genomic regions [4,5,6,7]
miR-22	↓TET2	HSC	provoke MDS in mice [106]	Significant prognostic value in MDS treated with HMAs [105]
Let-7 family	↓TET2	murine macrophages	↑ *IL-6* [107]	Altered methylation and expression profiles in MDS [10,11]
miR-101	↓EZH2	LSCC	↓autophagy, proliferation & ↑apoptosis [110]	Reduces incidence and delays the onset and progression of AML in mice [108]
miR-124	↓EZH2	MM	↓*CDKN2A* ↓proliferation &viability of myeloma cell line [111]	Potential marker of response to DAC in MDS/AML [109]
miR-21	↓EZH2 ↓DNMT1	Lung cancer SLE	↓*Cdc2, cyclinB1, BLC-2* [94] aberrant DNA hypomethylation [92]	Potential biomarker of epigenetic therapy in MDS [93]
miR-29b	↓DNTM3A/B, ↓DNMT1	AML	global DNA hypomethylation [96]	-
miR-29a	↓DNTM3A/B	Lung cancer	Re-expression of methylation-silenced tumor suppressor genes [97]	Downregulated in AML [98]
miR-221	↓DNTM3B	BC	↓ methylation levels of Nanog & Oct 3 [103]	Decreased levels in AML evolving from MDS [104]
miR-185	↓DNMT1	HCC	↑*PTEN*/Akt pathway	*PTEN* is associated with HMA resistance in MDS and progression to AML [79]
miR-152	↓DNMT1	PC	↑*PTEN*	*PTEN* is associated with HMA resistance in MDS and progression to AML [79]
miR-148a	↓DNMT1	SLE AML	aberrant DNA hypomethylation [92] ↑miR-148a, mutual negative feedback loop/↓cell proliferation & ↑ apoptosis [95]	-
miR-34b	↓DNMT1, HDAC1, HDAC2 & HDAC4	PC	↓proliferation through demethylation/active chromatin modifications/Akt pathway [112]	Cell viability inhibition/enhance cell apoptosis in AML cell lines [113]
miR-342	↓DNMT1	CRC	↓ cancer cell proliferation & invasion	Downregulation in AML [102]

↑ upregulated, ↓ downregulated, ↕ recruit, ↨ inhibit connection, Acute myeloid leukemia = AML, Small cell lung carcinoma = SCLC, Breast cancer = BC, Glioblastoma = GB, Colorectal cancer = CRC, Hematopoietic stem cell = HSC, laryngeal squamous cell carcinoma = LSCC, multiple myeloma = MM, Systemic Lupus Erythematosus = SLE, hepatocellular carcinoma = HCC, prostate cancer = PC.

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
