# Peer review of "Contingent Synergistic Interactions between Non-Coding RNAs and DNA-Modifying Enzymes in Myelodysplastic Syndromes"

_ijms, 2022, doi:10.3390/ijms232416069_

Round 1

Reviewer 1 Report

This review article by Symeonidis et al summarizes the current literature regarding the epigenetic alterations in MDS/AML with emphasis in the methylation patterns and the role of non-coding RNAs and microRNAs especially in cooperation with epigenetic regulators (methyltransferases and PRC2 complex components such as EZH2).

This is a really well-written review of the literature in a very interesting topic with research and clinical relevance in MDS and AML. I The authors have done an extensive literature search and have done a very comprehensive analysis/synthesis.  

I have only few minor points:

1. The authors should probably include a section in the introduction about non-coding RNAs and microRNAs. That way they can introduce the concept earlier to the reader.

2. It is mentioned in the introduction (line 82) that HMAs are used for patients with MDS who are not eligible for AlloBMT. That is not correct as most transplant centers will recommend treatment with a few cycles of HMAs before BMT. Thus, HMAs are useful even in patients who are eligible for BMT.

3. The AZA/DAC are introduced in the introduction. So no need to include the full names (Azacitidine and decitabine) in lines 182-183.

4. The authors mention that Response to HMA is unpredictable (line 202). That is not accurate. In clinical practice specific mutational patterns are associated with poor HMA response (https://pubmed.ncbi.nlm.nih.gov/25224413/, https://pubmed.ncbi.nlm.nih.gov/33591325/, https://pubmed.ncbi.nlm.nih.gov/30152885/). Moreover, a number of basic science studies have highlighted mechanisms of resistance to HMA, one of which is referenced by the authors (ref 78 in line 317). Thus, I would probably rephrase the comment that HMA response is unpredictable.

5. Given the new classifications of MDS (ICC, ELN) the borders between high-risk MDS and AML arising from MDS (sAML) are getting less prominent. Thus, the authors may consider adding a comment about how these findings/conclusions may be expanded for sAML (MDS/AML).  

Author Response

Our response:

We thank reviewer for helpful comments in the direction of improving our manuscript.

  1. We have included lines 70-73 within introduction section (in the resubmitted form of the manuscript with track changes), to highlight the relevance of non-coding RNAs differential expression in MDS, as requested.
  2. Lines 89-90 (in the resubmitted form of the manuscript with track changes) have been added following recommendation of the reviewer
  3. The words “azacytidine” and “decitabine” and the brackets have been removed (line 211 in the resubmitted form of the manuscript with track changes), according to recommendation.
  4. Text has been modified (lines 231-232 in the resubmitted form of the manuscript with track changes), according to recommendation of the reviewer.
  5. Text has been modified (lines 452-456 in the resubmitted form of the manuscript with track changes), according to recommendation of the reviewer.

Reviewer 2 Report

ijms-2080428

Contingent synergistic interactions between non-coding RNAs and DNA-modifying enzymes in Myelodysplastic Syndromes

Argiris Symeonidis, Theodora Chatziligeroudi, Vasiliki Chondrou, Argyro Sgourou

This is a well written review article about Myelodysplacia and non-coding RNAs. Since many mutations were reported in both splicing regulators and epigenetics factors, this manuscript is quite informative and useful for both basic scientists and clinicians.

I have some minor comments described below.

1) It would be helpful if the authors could prepare Figures to show schematic representations of DNA modifications and their modifying enzymes with possible outcomes.

2) Most of lnRNAs and miRNAs undergo RNA processing steps. It would be nicer if the authors could refer those steps.

3) There are some typos and missing spaces in the text. Please edit the text carefully.

Author Response

Our response:

We thank reviewer for helpful comments.

  1. Illustrations of epigenetic modifying enzymes and relevant modifications on chromatin have already been presented by others (Loaeza-Loaeza J. et al, 2020, Hillyar C. RT et al, 2022 ect), thereby we haven’t considered it as an apparent requirement for the present review article.
  2. Among non-coding RNA species, lncRNAs and miRNAs’ implication in MDS is thoroughly discussed in paragraph 2. Lines 214-223 and 231-240 (in the initially submitted manuscript) roughly outline the processing steps for lncRNAs and miRNAs respectively.
  3. Manuscript has been carefully edited to correct grammatical and syntactical weaknesses.

Reviewer 3 Report

The authors highlighted the synergistic interactions between non-coding RNAs and DNA-modifying enzymes in myelodysplastic syndromes. The review is well organized and literature is up-to-date. However the length of some paragraphs leads the readers to do not be concentrated on the topic of the review. My suggestions are below:

-The paragraph 1. “The course of aberrant DNA methylation patterns in MDS” is pretty long. Use sub-paragraphs to indicate sentences on DNMT3 enzymes, hypomethylating agents etc.

-Keep the focus of the review on the actual synergism and highlight examples of such interplay.

-Figure 1 is not very representative of the interplay between gene mutations and altered epigenetics. The authors should make a figure to illustrate the point. Right now the Figure illustrates the hematopoietic tree with aberrant myeloid transformation.

Line 145: Change the word “deviating” in “aberrant”.

Author Response

Our response:

We thank reviewer for helpful comments. Specific points are addressed and discussed below. Also, the manuscript has been carefully edited to correct grammatical and syntactical weaknesses.

  1. Paragraphs 1, 2 and 3 are equally long, but each one is a self-contained unit that captures the theoretical frame and presents an overall picture of epigenetic mechanisms with the perspective of their co-operation.
  2. The co-operation of epigenetic factors is reflected in paragraph 3 and in figure 2. A gradual presentation is formed in a way that implicated epigenetic mechanisms in MDS are presented initially and the potential synergism between epigenetic mechanisms is introduced as a final logical corollary, which can be inferred from their distinctive functions and relevant but not extended yet, experimental results.
  3. The present review article mainly refers to mutations that affect the enzymatic activity of components related to epigenetic machinery. Assessing that especially MDS pathophysiology exhibits a definite modified epigenetic background and main treatment administered consists of DNA hypomethylating agents that affect the epigenome, it is a logical consequence that epigenetic mechanisms should be investigated in detail. In this context, figure 1, illustrates the effects of combined genetic and epigenetic aberrations in hematopoietic stem cells, which jointly affect their maturation and differentiation by guiding specific cell subpopulations into clonal expansion, while inhibiting the differentiation of other cell populations. Rough corrective changes have been made to figure 1 and its legend to emphasize the reviewer’s point.
  4. It has been changed accordingly.